# A global analysis of national cardiovascular disease control plans using a multi-agent artificial intelligence model

Hugh Pearson[1], Caleb J. Kumar[1], Che L. Reddy[1], Estella Rose LeBlanc[2], Rifat Atun[1,3,4,5]*, With the CVD Control Collaborative[¶]

1 Health Systems Innovation Lab, Department of Global Health and Population, Harvard T.H. Chan School of Public Health, Harvard University, Boston, Massachusetts, United States of America, 2 Harvard College, Harvard University, Cambridge, Massachusetts, United States of America, 3 Department of Health Policy and Management, Harvard T.H. Chan School of Public Health, Boston, Massachusetts, United States of America, 4 Department of Global Health and Social Medicine, Harvard Medical School, Harvard University, Boston, Massachusetts, United States of America, 5 Faculty of Medicine, Imperial College London, London, United Kingdom

¶ Membership of the CVD Control Collaborative is provided in the Acknowledgements.
* ratun@hsph.harvard.edu

## Abstract

Cardiovascular diseases cause nearly one-third of global deaths, yet standalone National Cardiovascular Disease Control Plans remain uncommon and inconsistently structured. We assessed the comprehensiveness of recent national plans using a validated health-systems framework and a multi-agent artificial intelligence model. We identified the most recent official plan for 45 countries from World Health Organisation and World Heart Federation repositories and government sources. We adapted a health-systems planning framework for cardiovascular disease and validated it through a two-stage expert consensus process involving 42 specialists from 28 countries, resulting in 11 elements and 69 sub-elements with standardised definitions and scoring criteria. Plans were analysed using a three-stage artificial intelligence pipeline that ingested documents, applied framework-based scoring, and performed automated validation checks. Sub-elements were scored on a 0–5 scale and summarised by element, World Health Organisation region, and World Bank income group. Overall comprehensiveness was low (median 1.20/5). Plans most consistently addressed strategic direction (median 2.80) and governance arrangements (2.14). Contextual assessment was deficient — threats (0.12) and opportunities (0.29) — as were performance specification elements, including objectives (0.50) and health system outcomes (0.67). The Western Pacific region scored highest (median 1.71) and Africa lowest (0.90), though scores remained below moderate levels across all regions. Income group pairwise comparisons were non-significant across all groups; given the small LIC sample (n = 2), no inferential conclusions about income group differences are drawn. Validation against blinded human review across six countries showed 43.7% exact agreement and 68.0% agreement within one point;

**Data availability statement:** The national cardiovascular disease control plans analysed in this study are publicly available through the World Health Organisation Noncommunicable Disease Document Repository (https://extranet. who.int/ncdccs/documents/) and government websites. The scoring dataset and human validation data have been deposited in the Harvard Dataverse and are available at https://dataverse. harvard.edu/dataverse/NCVDCP_Analysis.

**Funding:** This work was supported by Harvard University (to HP, CLR, ERL, RA). CK was supported by funding from the Virchow Foundation to the Global Health Policy Lab Initiative. The funders had no role in study design, data collection and analysis, decision to publish, or preparation of the manuscript.

**Competing interests:** I have read the journal's policy and the authors of this manuscript have the following competing interests: RA reports institutional grants from Novo Nordisk, the Virchow Foundation, and the Bill & Melinda Gates Foundation, and honoraria from Merck & Co, all unrelated to the submitted work. HP, CLR, CK, and ERL declare no competing interests.

ordinal agreement statistics were uniformly weak and non-significant, indicating the approach is validated for structural benchmarking rather than fine-grained qualitative judgement. Most national cardiovascular disease plans articulate vision without sufficient operational detail, particularly for contextual analysis, measurement, and integrated financing. Standardised planning templates and artificial intelligence–supported benchmarking, complemented by expert review, could strengthen national planning quality and enable scalable global comparisons.

## Author summary

Cardiovascular disease remains the world's leading cause of death, yet the national strategies designed to fight it often lag behind those for other major illnesses like cancer. In this study, we set out to understand the quality of these strategies by analysing 45 national control plans from around the globe. Using a specialised artificial intelligence tool validated against independent human review, we evaluated how comprehensive these plans truly are against a standard health-system framework. We found that while most countries are effective at setting high-level goals and identifying leadership structures, they frequently fail to include the practical details necessary for implementation, such as specific budgets, local risk analysis, and clear methods to measure progress. Plan quality did not differ significantly across income groups in this sample, suggesting that the global deficit in planning may reflect the absence of standardised methodology more than resource constraints alone. Our findings highlight an urgent need for evidence-based planning guides to help governments transition from political promises to practical action. We also demonstrate that while artificial intelligence can efficiently analyse large volumes of health policy, it serves best as a partner to human judgment rather than a replacement.

## Introduction

Cardiovascular diseases (CVDs) are the leading cause of death globally, accounting for an estimated 19.8 million deaths annually and 32% of all global mortality in 2023 [1–3]. Although this burden is widespread, differences in demographics, economic conditions, and health-system capacity create substantial disparities in how CVDs are prevented, diagnosed, treated, cared for, and rehabilitated worldwide [4,5].

National Cardiovascular Disease Control Plans (NCVDCPs) are policy instruments used to organise the prevention, treatment, and rehabilitation of CVD [6,7]. Implementing these integrated, system-wide plans is central to fulfilling the World Health Assembly's WHA70.12 resolution (2017), which calls on Member States to adopt coordinated approaches to non-communicable disease (NCD) control that set clear objectives, indicators, and monitoring arrangements, and that strengthen financing, workforce, and access to essential medicines with an explicit equity focus

[8]. NCVDCPs are also critical to achieving the UN Sustainable Development Goal target of a 30% reduction in premature mortality from NCDs by 2030 [9].

However, national strategic planning for CVD lags behind other major disease areas. Research by the World Heart Federation (WHF) indicates that while 87% of countries have national cancer control plans, only 7% have developed standalone strategies for CVD [7]. Where CVD plans do exist, there is no globally agreed or standardised format to guide their structure, comprehensiveness or quality, despite roadmaps published by the WHF and WHO [6,10].

Large language models (LLMs) offer a scalable approach to policy appraisal, enabling rapid, consistent, high-granularity analysis of complex documents while mitigating the resource constraints and subjectivity inherent in traditional manual reviews [11,12]. To address current gaps in CVD strategic planning, we applied artificial intelligence-based methods with human oversight embedded at every stage of development and assessment [13,14], to evaluate the comprehensiveness of a global sample of NCVDCPs – drawn predominantly from high-income and European settings. Applying this framework, we compared comprehensiveness across country income groups and WHO regions to identify opportunities to strengthen national CVD planning.

## Methods

### National cardiovascular disease control plan collection

We identified the most recent official NCVDCP for each country by searching the WHO NCD Document Repository [15], WHF resources [6], and government websites. Inclusion was limited to CVD specific national plans in any language. We excluded broader NCD plans with CVD components, sub-national plans, draft documents, and superseded versions.

### Analytical framework development

To define the foundational content of a comprehensive plan, we adapted the 2008 health-systems framework by Atun et al. [16]. Originally used to analyse European National Cancer Control Plans (NCCPs), this framework takes a systems perspective to define the elements of an 'ideal' plan and is designed for scalable application across contexts. We recontextualised this model for CVD by undertaking a literature review identifying current approaches to NCVDCP development and by incorporating guidance from the WHF CVD Roadmaps [6] and the WHO HEARTS Technical Package [10]. The resulting template comprises 11 elements and 69 sub-elements, each with standardised definitions, indicators, and scoring criteria specific to NCVDCPs.

Sub-elements were scored on a 0–5 ordinal scale with criteria defined prospectively in the analytical framework: 0 = not discussed; 1 = partially discussed, no recommended indicators used; 2 = partially discussed with some recommended indicators; 3 = fully addressed with some or all recommended indicators; 4 = fully addressed with all recommended indicators and national baseline data; 5 = fully addressed with all indicators, baseline data, and specific measurable targets. Scoring criteria were sub-element-specific and applied identically by PRISM and human reviewers; full criteria are provided in the supplementary information (S1 Table).

### Analytical framework validation

The analytical framework was validated through a modified two-stage Delphi process [17]. The expert panel, the CVD control collaborative, comprised 42 specialists in clinical cardiology, health policy, public health, and health systems recruited from 28 countries via the networks of the WHF, Global Health Policy Lab, Wellcome Trust/India Alliance, and the Harvard Health Systems Innovation Lab.

In the first stage (August to October 2025), participants completed an online questionnaire to rate the relevance of sub-elements on a 1–5 scale, assess definition accuracy, and suggest additions to definitions and indicators. Initial ratings indicated moderate-to-high relevance (overall mean 3.90/5). Elements scoring below 4 – Health System Performance

Outcomes (3.40), Outputs (3.42), Objectives (3.44), Threats (3.79), and Opportunities (3.87) – were prioritised for detailed revision in the subsequent stage (S2 Text). Qualitative feedback from stage one prompted substantial refinements to the framework, particularly regarding the integration of comprehensive care models and value-based metrics (S3 Text). Definitions were expanded to explicitly include the management of key comorbidities, such as periodontal disease, and to incorporate patient-reported outcome measures that capture wellbeing alongside traditional epidemiological data. The revised framework also added specific indicators for vulnerable populations, multidisciplinary workforce integration, and high-impact, low-cost interventions such as polypills and sodium reduction strategies.

During stage two, conducted on October 23, 2025, a subset of 16 experts from the original 42 convened in a virtual roundtable to finalise the framework. Revisions included nine major content additions, such as metrics for pregnancy-related CVD risk, gender equity, and economic productivity gains. Furthermore, definitions were refined to emphasise primary care–led hypertension management, community-based delivery models, and digital health integration (S4 Text). The final framework shown in Table 1 encompasses 11 elements and 69 sub-elements, requiring the assessment of system performance, contextual threats, and opportunities; the definition of strategy; the specification of governance, financing, and resource reforms; the delineation of service interventions; and a comprehensive implementation plan.

## Policy Reasoning Integrated Sequential Model Development (PRISM)

To facilitate rapid, consistent and scalable analysis, we developed the Policy Reasoning Integrated Sequential Model (PRISM), a three-stage multi-agent system (S1 Text). Consistent with human-in-the-loop principles, domain experts were embedded at each stage of the system's lifecycle—informing framework design through the Delphi process, guiding prompt engineering and model selection, and validating outputs against independent human review [13,14]. In this context, an agent is defined as an autonomous AI model executing designated tasks within a coordinated workflow. In stage one, a document-ingestion agent (Qwen2.5-VL-72B) processed policy documents of heterogeneous format and layout. This agent applied automated preprocessing to mitigate noise and geometric distortion, utilising spatial-aware text extraction with integrated visual recognition to digitise and structure the content [18]. In stage two, a policy-analysis agent (Llama 4 Scout 70B) applied the 69 sub-elements of the analytical framework to the NCVDCPs. Scoring criteria required only that relevant content be discussed; implicit or narrative treatment without explicit headings was eligible for non-zero scores. By employing retrieval-augmented generation (RAG) and structured prompts, this agent produced standardised, machine-readable Javascript Object Notation (JSON) outputs [11,12]. Finally, in stage three, a semantic-validation agent assessed semantic coherence and JSON-schema conformity, triggering threshold-based re-analysis of flagged sections to yield the final analytic dataset.

## Policy reasoning integrated sequential model validation

We assessed PRISM's accuracy of analysis and concordance against independent human review, consistent with human-in-the loop-principles. Six countries were purposively selected to reflect variation in programme maturity, region, and income group – Ghana, India, Myanmar, Turkey, United Kingdom, and the United States. Three human experts blinded to PRISM outputs independently scored a subset of plans using the same analytical framework. PRISM evaluated the same plans with identical criteria. Each reviewer was assigned two countries — Ghana and India (Reviewer 1), Myanmar and the United States (Reviewer 2), Turkey and the United Kingdom (Reviewer 3) — with no shared observations across reviewers. Scores were compared using Spearman ρ, linear- and quadratic-weighted κ, ICC (3,1) as ordinal-appropriate statistics, alongside exact and within-1-point agreement rates.

PRISM showed variable concordance with human ratings across 403 matched sub-element comparisons. Exact agreement occurred in 43.7% of comparisons (176/403), and 68.0% were within one point (274/403). Ordinal agreement statistics were uniformly weak and non-significant (Spearman ρ = 0.061, p = 0.222; linear-weighted κ = 0.049; quadratic-weighted κ = 0.045; ICC (3,1) = 0.046, p = 0.181; Pearson r = 0.048, p = 0.34, reported for reference only). A floor effect was present:

**Table 1. Framework to analyse National Cardiovascular Disease Control Plans: Description of the 11 elements and 69 sub-elements by health system theme.**

| | | |
|---|---|---|
| **Current Health System Performance in relation to CVD control** | **ELEMENT 1: HEALTH SYSTEM PERFORMANCE OUTCOMES** | |
| | **How well is the health system performing in relation to care and control concerning health system outcomes for cardiovascular disease?** | |
| | Health | How the health system is currently performing in relation to cardiovascular disease (CVD), assessed in terms of health outcomes. |
| | Financial Risk Protection | How the health system is currently performing in relation to CVD, assessed in terms of financial risk protection outcomes. |
| | User Satisfaction | How the health system is currently performing in relation to CVD, assessed in terms of user satisfaction. |
| | **ELEMENT 2: HEALTH SYSTEM PERFORMANCE OBJECTIVES** | |
| | **How well is the health system producing health services for CVD in terms of the four value dimensions?** | |
| | Effectiveness | How the health system is currently performing in relation to CVD, assessed in terms of the delivery of health services that are effective at improving health. |
| | Efficiency | How the health system is currently performing in relation to CVD, assessed in terms of the delivery of health services that are efficient, maximizing health outcomes relative to the resources used. |
| | Equity | How the health system is currently performing in relation to CVD, assessed in terms of the delivery of health services that are equitable, ensuring that access, quality of care, and outcomes do not vary across population subgroups. |
| | Responsiveness | How the health system is currently performing in relation to CVD, assessed in terms of its ability to meet the legitimate non-health expectations of users. |
| | **ELEMENT 3: HEALTH SYSTEM PERFORMANCE OUTPUTS** | |
| | **Is the health system delivering the right health services to respond to CVD?** | |
| | Individual Health Services | Describes the performance of health system outputs, assessed in terms of the delivery, integration, and care model of individual health services for CVD provided to patients at primary, secondary, and tertiary levels. |
| | Population Health Services | Describes the performance of health system outputs, assessed in terms of the delivery of population-level health services for CVD prevention and control. |
| | Community-Based Palliative Care | Assesses the delivery and integration of community-based palliative and supportive care services for patients with advanced CVD. |
| | CVD Surveillance Systems | How the health system is currently performing in relation to CVD, assessed in terms of the establishment and operation of CVD surveillance systems. |
| **Contextual Forces influencing CVD Control** | **ELEMENT 4: HEALTH SYSTEM THREATS** | |
| | **What contextual threats will influence CVD demands on the health system and influence the system response to CVD care and control?** | |
| | Demographic | Demographic factors that pose a threat to CVD control. |
| | Epidemiological | Epidemiological factors that pose a threat to CVD control. |
| | Political | Political factors that pose a threat to CVD control. |
| | Legal | The absence of a supportive legal and regulatory framework that could mitigate CVD risk factors and strengthen the health system's response. |
| | Sociocultural | Sociocultural factors that pose a threat to CVD control. |
| | Economic | Economic factors that pose a threat to CVD control. |
| | Ecological | Ecological and environmental factors that pose a threat to CVD control. |
| | Technological | Technological factors that pose a threat to CVD control. |
| | **ELEMENT 5: HEALTH SYSTEM OPPORTUNITIES** | |
| | **What contextual opportunities will influence CVD demands on the health system and influence the system response to CVD care and control?** | |
| | Demographic | Demographic factors that present an opportunity for CVD control. |
| | Epidemiological | Epidemiological factors that present an opportunity for CVD control. |
| | Political | Political factors that present an opportunity for CVD control. |
| | Legal | The existence of a strong legal and regulatory framework that enables public health action, promotes healthy environments, and ensures access to care. |
| | Sociocultural | Sociocultural factors that present an opportunity for CVD control. |
| | Economic | Economic factors that present an opportunity for CVD control. |
| | Ecological | Ecological and environmental factors that present an opportunity for CVD control. |
| | Technological | Technological factors that present an opportunity for CVD control. |

*(Continued)*

**Table 1.** (Continued)

| CVD Strategy | **ELEMENT 6: CVD STRATEGY** | |
|---|---|---|
| | **How will the challenges identified be addressed to improve health system performance and create value in relation to CVD?** | |
| | Vision | A forward-looking statement that describes what the plan aspires to achieve in the future, grounded in a comprehensive needs assessment. |
| | Mission | A statement that defines the plan's actionable purpose and what it will do for the health systems, its users, citizens and society supported by a clearly defined governance structure |
| | Goals | The high-level outcomes the plan aims to achieve in terms of health outcomes, financial risk protection, and user satisfaction, aligned with the mission and vision. |
| | Objectives | How value will be created in terms of efficiency, effectiveness, equity, and responsiveness to support the attainment of the plan's goals. |
| | Values | Core beliefs and principles that should guide stakeholder behaviour, decisions, and relationships when implementing the plan, reflecting the overall culture that defines actions within the health system. |
| | **ELEMENT 7: GOVERNANCE AND ORGANISATION** | |
| | **What reforms are to be undertaken in terms of governance and organisation of the health system to achieve the articulated strategy to improve CVD care and control?** | |
| | Macro-organisation | Key organisations within the Ministry of Health and other relevant government or quasi-governmental macro-level organisations that will be established or currently existing that assume the responsibility for overseeing the implementation and monitoring of the National CVD Plan. |
| | Governance | How the principles of accountability and transparency will be upheld concerning plan implementation. |
| | Policy | List and description of other relevant policies at the domestic level and resolutions or guidelines at the global level that influence the National CVD Plan. |
| Proposed interventions in relation to system functions and outputs | Regulation | Regulations that will be established to strengthen CVD care and control. |
| | Decentralization | Reforms implemented to change the extent of decentralization within the health system to enable plan implementation. |
| | Strategic public private partnerships | The design and implementation of strategic PPPs that could be used to expand funding, align incentives of key stakeholders, and promote the achievement of priority activities and outcomes in the National CVD Plan. |
| | Integration with Noncommunicable Disease Programs | Strategies to integrate CVD control efforts with broader noncommunicable disease prevention and control programs. |
| | **ELEMENT 8: FINANCING** | |
| | **What reforms are to be undertaken in terms of financing of the health system to achieve the articulated strategy to improve CVD care and control?** | |
| | Cost measurement systems | Practices and efforts to ascertain the cost of delivering health services for CVD care and control, including both direct medical costs and indirect societal costs. |
| | Current financing and fiscal space | Current level of funding allocated to health services for CVD. |
| | Proposed funding to implement plan | The estimated funding range or detailed budget required to successfully implement the National CVD Plan. |
| | Sources of funds | New sources of funding, including innovative financing mechanisms, that will be designed and introduced to expand fiscal space for CVD care and control. |
| | Pooling of funds | How funds from different sources will be pooled at national or sub-national levels to purchase CVD services and reduce financial barriers for patients. |
| | Channelling of funds | The entities that will receive funding to implement activities prioritized by the National CVD Plan. |
| | Allocation of funds | The amount of, and rationale for, fund allocation in relation to the specific health system resources and activities listed and described in the plan's budget. |
| | Payment mechanisms for providers | How healthcare providers will be remunerated for delivering CVD services. |
| | Payment mechanisms for capital investments | How procurement processes will be executed with an emphasis on transitioning towards value-based methods of procurement. |
| | Resource Mobilization Strategies | Approaches to secure and sustain financial and non-financial resources for NCVDCP implementation. |

*(Continued)*

| | |
|---|---|
| | **ELEMENT 9: RESOURCE MANAGEMENT** |
| | **What reforms are to be undertaken in terms of resource management of the health system to achieve the articulated strategy to improve CVD care and control?** |
| Human resources | The training, recruitment, interprofessional integration, equitable distribution, and deployment of a qualified health workforce to prevent, manage, and rehabilitate patients with cardiovascular conditions. |
| Infrastructure | Facilities, equipment and devices, and the maintenance of these facilities and equipment |
| Pharmaceuticals and medical supplies | The essential medicines, therapeutics, and supplies used for the prevention, diagnosis, and treatment of cardiovascular disease. |
| Information technology and data systems | Encompass the infrastructure and processes required for data systems to inform outcome measurements, clinical decision-making, research, innovation, and health policy and planning. |
| Supply chain management | The procurement, storage, distribution, inventory management, and monitoring of essential medical products, equipment, and pharmaceuticals necessary for CVD prevention, diagnosis, and treatment. |
| Research | Efforts designed to generate new knowledge in relation to cardiovascular disease care and control. |
| Innovation ecosystem | The institutionalisation of push and pull strategies and policy changes that promote emergent and driven innovation to encourage the design, development, implementation, and scale-up of new solutions for CVD care. |
| Education Initiatives | How the health system is currently performing in relation to CVD, assessed in terms of educational initiatives that promote CVD awareness, prevention, and early detection among the population. |
| Innovation Initiatives | Strategies and initiatives to enhance the skills, knowledge, and availability of health professionals involved in CVD control. |
| | **ELEMENT 10: HEALTH SERVICES** |
| | **What reforms are to be undertaken in terms of the delivery of health services to achieve the articulated strategy to improve CVD care and control?** |
| Public Health Services (Health promotion) | Primary prevention interventions focusing on the social and environmental determinants of CVD that empower individuals to lead healthier lifestyles in relation to CVD risk factors. |
| Public Health Services (Health protection) | Primary prevention interventions focusing on environmental and policy determinants to protect individuals and populations from CVD. |
| Public Health Services (Disease Prevention) | Interventions aimed at preventing CVD and promoting early detection through screening for key risk factors. |
| Personal Healthcare Services (Diagnosis) | The personal healthcare services to be delivered to enhance CVD diagnosis. |
| Personal Healthcare Services (Treatment) | The personal healthcare services to be delivered to enhance CVD treatment. |
| Personal Healthcare Services (Palliation and rehabilitative care) | The personal healthcare services to be delivered to enhance rehabilitation and palliation for individuals with CVD. |
| Provider Value Enhancement | Measures to be taken by providers to improve value in the delivery of health services. |
| **Translation** | **ELEMENT 11: MONITORING AND EVALUATION** |
| | **How will the plan be implemented, measured, stakeholder alignment and sustained change management be achieved?** |
| Monitoring and Evaluation framework | How the achievement of the plan is evaluated in relation to its inputs, activities, outputs, and outcomes. |
| Change management | How change management will be achieved and the approach that will be pursued to promote transparency, accountability, and the adoption of new practices by stakeholders. |
| Risk and Mitigation Strategies | The identification and analysis of major risks that could influence the implementation of the cardiovascular disease control plan including political risks, financial, and operational risks, and the mitigation measures that will be pursued to better manage these risks. |
| Stakeholder engagement | The expectations, responsibilities, and plan for engaging key stakeholders in the implementation of the plan, including other government ministries, sub-national governments, the private sector, non-governmental organisations, professional societies, and health professionals themselves. |

61.0% of LLM scores and 62.3% of human scores were 0, inflating per-element exact agreement for elements with sparse policy content. Within one-point agreement by country ranged from 75.0% (Ghana, United States), to 45.2% (India), with the United Kingdom at 73.1%, Turkey at 69.6% and Myanmar at 68.1%. Bias was bidirectional and score-dependent: PRISM over-scored relative to human reviewers when expert scores were 0 (mean difference +0.96) but under-scored when expert scores were $\geq 2$ (mean differences −0.93 to −4.00). Overall mean difference was +0.28 points. Collapsing scores into three tiers, PRISM over-scored at low human ratings (scores 0–1, n = 318, mean difference +0.75) and under-scored at medium (scores 2–3, n = 72, mean difference −1.22) and high ratings (scores 4–5, n = 13, mean difference −3.08). As each reviewer assessed a distinct pair of countries with no overlapping observations, formal inter-rater reliability could not be assessed.

### Policy reasoning integrated sequential model application

We analysed NCVDCPs on an Amazon Web Services Elastic Compute Cloud Instance, generating standardised scores for all 69 sub-elements with automated quality assurance (S1 Text).

### Comparative analysis

Following the application of PRISM, we analysed NCVDCP comprehensiveness within and across World Bank country income groups and WHO country regions.

In this context, comprehensiveness is defined as the extent to which a plan addresses the 11 elements and 69 sub-elements of the validated framework, reflecting both the breadth of topic coverage and the depth of specific detail. Non-zero scores reflect the documented presence and specificity of content within the plan; they do not indicate the adequacy, effectiveness, or implementation success of the policies described. Sub-element scores are ordinal; numerical aggregates (medians, means, IQRs) are reported as heuristic indicators of directional patterns across plans and should not be interpreted as precise quantitative distances between rubric levels. Data processing and visualisation were conducted in Python, utilising the pandas, numpy, and matplotlib libraries. Scores for the 69 sub-elements were normalised to a standard 0–5 scale. Element and country-level scores were derived by taking the median across constituent sub-elements, applying equal weight to each sub-element; this treats breadth of coverage within an element as substantively meaningful. Median rather than mean was used as the primary summary statistic, consistent with the ordinal scale. A sensitivity analysis comparing sub-element-equal weighting against element-equal weighting and mean against median aggregation confirmed that principal findings were robust across approaches (S2 Table). Non-parametric Mann-Whitney U tests were used for income group comparisons.

## Results

### Plan characteristics

We identified 45 NCVDCPs outlined in Table 2. The majority originated from high-income countries (n = 27), followed by upper-middle-income (n = 10), lower-middle-income (n = 6), and low-income countries (n = 2). Geographically, the plans were predominantly from the European region (n = 22, 48.9%), with limited representation from the Americas (n = 8), Africa (n = 4), the Western Pacific (n = 4), the Eastern Mediterranean (n = 5), and South-East Asia (n = 2). This distribution reflects the global scarcity of standalone CVD plans and limits generalisability beyond high-income, European settings.

### Overall comprehensiveness

The median overall comprehensiveness score across all plans was 1.20/5, as shown in Table 3, indicating that most plans address framework elements only partially, without specific indicators or baseline data. Ten of the eleven elements scored below 2.0. Contextual analysis represented the most critical gap, with negligible scores for threats (median 0.12) and

**Table 2. List of National Cardiovascular Disease Plans and broader policy landscape by country.**

| Country | Plan Title | NCVDP Year | NCDP Year | UHCP Year | NHSP Year | FCTC Year |
|---|---|---|---|---|---|---|
| Algeria | Programme National de Lutte | 2000 | 2015 | 1975 | 2010-2014 | 2006 |
| Argentina | National program for the prevention of cardiovascular diseases | 2011 | 2019 | 2016 | 2022–2025 | — (signed 2003, not ratified) |
| Australia | National strategic action plan for heart disease and stroke | 2020 | 2021 | 1984 | 2020–2025 | 2004 |
| Canada | Heart Health Strategy and Action Plan | 2009 | 2018 | 1984 | 2021–2025 | 2004 |
| Chile | Technical guidance for the cardiovascular health program | 2017 | 2022 | 2005 | 2022–2030 | 2005 |
| Costa Rica | Analysis and planning for the approach to comprehensive care of cardiovascular pathology at the national level | 2016 | 2014 | 1941 | 2023–2033 | 2008 |
| Cuba | National program for prevention, diagnosis, evaluation, treatment and rehabilitation of ischemic heart disease | 2015 | 2019 | 1983 | 2022–2030 | — (signed 2004, not ratified) |
| Czechia | National cardiovascular program of the Czech Republic | 2013 | 2020 | 1997 | 2020–2030 | 2012 |
| Ecuador | Operational guidelines for the implementation of the HEARTS initiative in Ecuador | 2021 | 2013 | 2008 | 2022–2031 | 2006 |
| Estonia | National strategy for the prevention of cardiovascular diseases | 2005 | 2020 | 2002 | 2020–2030 | 2005 |
| Germany | Healthy Heart Act - GHG | 2024 | 2015 | 1883 | 2021–2025 | 2004 |
| Ghana | National guidelines for the management of cardiovascular diseases | 2024 | 2022 | 2003 | 2022–2025 | 2004 |
| Greece | National action plan for cardiovascular diseases | 2008 | 2019 | 1983 | 2021–2025 | 2006 |
| Hungary | National programme for preventing and treating cardiovascular diseases | 2006 | 2018 | 1990 | 2021–2030 | 2004 |
| India | National multisectoral action plan for prevention and control of common non-communicable diseases | 2017 | 2017 | 2018 | 2017–2030 | 2004 |
| Ireland | Changing cardiovascular health: national cardiovascular health policy | 2010 | 2017 | 2017 | 2021–2025 | 2005 |
| Italy | Epidemiology and prevention of cardio-cerebrovascular diseases | 2011 | 2020 | 1978 | 2019–2021 (extended) | 2008 |
| Japan | The National Plan for Promotion of Measures Against Cerebrovascular and Cardiovascular Diseases | 2020 | 2013 | 1961 | 2024–2035 | 2004 |
| Kenya | Kenya national guidelines for cardiovascular diseases management | 2018 | 2021 | 2023 | 2023–2027 | 2004 |
| Latvia | Cardiovascular Health Improvement Action Plan | 2013 | 2021 | 2018 | 2021–2027 | 2005 |
| Lithuania | Description of the direction of reducing morbidity and premature mortality from cerebrovascular diseases | 2015 | 2014 | 1996 | 2014–2025 | 2004 |
| Luxembourg | National Cardio-Neuro-Vascular Disease Plan | 2022 | 2019 | 2010 | 2023–2028 | 2005 |
| Mexico | Specific Action Program: Cardiometabolic Diseases | 2019 | 2020 | 2022 | 2020–2024 | 2004 |
| Morocco | Specific Action Program: Cardiometabolic Diseases | 2018 | 2019 | 2022 | 2018–2025 | — (signed, not ratified) |
| Myanmar | National Strategic Plan For Cardiovascular Disease Services | 2019 | 2017 | 2017 | 2017–2021 | 2004 |
| New Zealand | Targeting Diabetes and Cardiovascular Disease | 2011 | 2020 | 1938 | 2023– | 2004 |
| Portugal | National Program for Cerebro-Cardiovascular Diseases | 2017 | 2013 | 1979 | 2021–2030 | 2005 |
| Qatar | National Cardiovascular Disease Prevention Strategy | 2018 | 2017 | 2022 | 2018–2022 | 2004 |
| Republic of Korea | Comprehensive Plan for cardiovascular diseases | 2018 | 2021 | 2000 | 2023–2030 | 2006 |

*(Continued)*

**Table 2.** (Continued)

| Country | Plan Title | NCVDP Year | NCDP Year | UHCP Year | NHSP Year | FCTC Year |
|---|---|---|---|---|---|---|
| Republic of Moldova | National Program for Prevention and Control of Cardiovascular Diseases | 2014 | 2016 | 2004 | 2023–2030 | 2009 |
| Romania | National Strategy for Combating Cardiovascular and Cerebrovascular Diseases | 2024 | 2014 | 1999 | 2023–2030 | 2006 |
| Russian Federation | Fight against cardiovascular diseases | 2019 | 2019 | 1993 | 2019–2024 | 2008 |
| Saudi Arabia | Strategy for the cardiovascular disease control program | 2019 | 2019 | 2019 | 2019– | 2005 |
| Senegal | Operational plan against cardiovascular and metabolic diseases | 2016 | 2017 | 2013 | 2019–2028 | 2005 |
| Serbia | National Program for Prevention, Treatment and Control of Cardiovascular Diseases in the Republic of Serbia | 2010 | 2019 | 2019 | 2018–2026 | 2006 |
| Slovenia | Rules for carrying out preventive cardiovascular health care at the primary level | 2015 | 2020 | 1992 | 2016–2025 | 2005 |
| Spain | Stroke strategy of the national health system | 2009 | 2022 | 1986 | 2022– | 2005 |
| Switzerland | National strategy against cardiovascular diseases, stroke and diabetes | 2016 | 2017 | 1996 | 2020–2030 | — (signed, not ratified) |
| Tajikistan | Assessment and pilot introduction of essential interventions for the management of hypertension and cardiovascular diseases at the primary health care level in the Republic of Tajikistan | 2018 | 2018 | 2021 | 2021–2030 | 2013 |
| Trinidad and Tobago | Trinidad and Tobago HEARTS Initiative (TTHI) Implementation Plan | 2019 | 2017 | 2023 | 2023 | 2004 |
| Turkey | Prevention and Control Program for Cardiovascular Diseases: Strategic Plan and Action Plan for the Risk Factors | 2009 | 2017 | 2012 | 2019–2023 | 2004 |
| Turkmenistan | Concept for the prevention of morbidity, mortality and disability from cardiovascular diseases | 2017 | 2016 | 1995 | 2021–2025 | 2011 |
| United Arab Emirates | Unified policy for reducing mortality from cardiovascular diseases | 2027 | 2017 | 2014 | 2023–2026 | 2005 |
| United Kingdom | Public Health England cardiovascular disease prevention initiatives | 2018 | 2019 | 1948 | 2019–2029 | 2004 |
| United States | Charting the Future Together: The NHLBI Strategic Vision | 2016 | 2011 | 2010 | 2020-2030 | (signed 2004, not ratified) |

**Notes:** NCVDP: National Cardiovascular Disease Plan; NCDP: Non-Communicable Disease Plan, Universal Health Coverage Plan; NHSP = National Health Strategic Plan, FCTC = Framework Convention on Tobacco Control

opportunities (0.29); elements assessing current health system performance were also consistently under-specified, including objectives (0.50), outcomes (0.67), and outputs (0.75). Conversely, CVD strategy (2.80) and governance and organisation (2.14) were the most comprehensively addressed elements. Implementation/monitoring and evaluation (1.75), financing (1.44), and health services (1.43) were mentioned but lacked the specificity, indicators, or baseline data required for substantive coverage.

## Regional variation

Comprehensiveness varied markedly across regions as shown in Fig 1. Regional patterns are presented descriptively. Findings for South-East Asia (n = 2) and Africa and Western Pacific (n = 4 each) should be interpreted with caution given

Table 3. Comprehensiveness scores for the National Cardiovascular Disease Plans by World Bank income group and World Health Organisation region.

| | Element 1 Outcomes | Element 2 Objectives | Element 3 Outputs | Element 4 Threats | Element 5 Opportunities | Element 6 CVD Strategy | Element 7 Governance and Organisation | Element 8 Financing | Element 9 Resource Management | Element 10 Health Services | Element 11 Monitoring and Evaluation | TOTAL |
|---|---|---|---|---|---|---|---|---|---|---|---|---|
| All countries (n=45) | 0.67 [0.33, 0.75] | 0.50 [0.25, 1.00] | 0.75 [0.50, 1.00] | 0.12 [0.12, 0.29] | 0.29 [0.12, 0.50] | 2.80 [2.20, 3.20] | 2.14 [1.43, 2.43] | 1.44 [1.00, 2.00] | 1.22 [0.67, 1.88] | 1.43 [1.00, 1.86] | 1.75 [0.50, 2.50] | 1.20 [0.92, 1.49] |
| LIC (n=2) | 0.50 [0.42, 0.58] | 0.88 [0.81, 0.94] | 0.75 [0.62, 0.88] | 0.31 [0.22, 0.41] | 0.38 [0.38, 0.38] | 2.50 [2.15, 2.85] | 2.64 [2.04, 3.25] | 2.55 [2.48, 2.62] | 1.39 [1.31, 1.47] | 1.43 [1.36, 1.50] | 2.00 [1.50, 2.50] | 1.39 [1.32, 1.46] |
| LMIC (n=6) | 0.50 [0.33, 0.67] | 0.62 [0.31, 1.50] | 0.50 [0.50, 0.69] | 0.00 [0.00, 0.19] | 0.27 [0.04, 0.47] | 2.60 [1.75, 3.00] | 1.60 [1.42, 1.80] | 1.25 [0.82, 1.30] | 0.76 [0.57, 1.35] | 0.86 [0.71, 1.47] | 0.38 [0.06, 1.06] | 0.90 [0.72, 1.14] |
| UMIC (n=10) | 0.58 [0.27, 0.92] | 0.38 [0.25, 0.69] | 0.71 [0.50, 1.00] | 0.13 [0.12, 0.34] | 0.12 [0.12, 0.28] | 2.40 [2.10, 2.95] | 1.57 [1.29, 2.38] | 1.50 [0.93, 1.90] | 1.94 [0.92, 2.08] | 1.31 [0.94, 1.68] | 1.25 [0.56, 2.19] | 1.13 [0.90, 1.37] |
| HIC (n=27) | 0.67 [0.42, 0.88] | 0.75 [0.25, 1.12] | 0.75 [0.67, 1.00] | 0.12 [0.12, 0.33] | 0.38 [0.25, 0.71] | 3.00 [2.40, 3.20] | 2.29 [1.79, 2.43] | 1.50 [1.10, 1.95] | 1.22 [0.70, 1.67] | 1.50 [1.14, 2.29] | 2.00 [0.75, 2.50] | 1.30 [1.05, 1.67] |
| Africa (n=4) | 0.67 [0.50, 0.83] | 1.12 [0.44, 1.81] | 0.62 [0.50, 0.98] | 0.00 [0.00, 0.06] | 0.27 [0.12, 0.41] | 2.30 [1.50, 3.05] | 1.47 [1.32, 1.64] | 0.95 [0.67, 1.23] | 0.76 [0.61, 1.04] | 0.86 [0.71, 1.21] | 0.38 [0.19, 0.69] | 0.90 [0.80, 1.02] |
| Americas (n=8) | 0.67 [0.44, 0.67] | 0.25 [0.25, 0.81] | 1.00 [0.62, 1.10] | 0.12 [0.12, 0.17] | 0.19 [0.12, 0.38] | 3.00 [2.70, 3.20] | 1.91 [1.50, 2.57] | 1.60 [1.08, 1.82] | 1.66 [1.13, 2.05] | 1.50 [1.14, 1.89] | 2.25 [1.31, 2.50] | 1.26 [1.11, 1.55] |
| Eastern Mediterranean (n=5) | 0.67 [0.33, 0.75] | 0.75 [0.00, 0.75] | 0.75 [0.50, 1.25] | 0.25 [0.12, 0.29] | 0.60 [0.25, 0.67] | 3.00 [2.80, 3.00] | 2.43 [2.29, 2.71] | 2.10 [1.90, 2.10] | 1.75 [1.56, 1.78] | 2.38 [1.62, 2.57] | 2.33 [2.25, 2.50] | 1.46 [1.41, 1.79] |
| Europe (n=22) | 0.67 [0.33, 0.94] | 0.62 [0.31, 1.00] | 0.67 [0.50, 1.00] | 0.12 [0.12, 0.34] | 0.31 [0.13, 0.42] | 2.40 [2.00, 3.15] | 2.29 [1.43, 2.43] | 1.35 [0.93, 1.70] | 1.06 [0.56, 1.50] | 1.33 [0.82, 1.68] | 1.38 [0.54, 2.25] | 1.14 [0.87, 1.40] |
| South-East Asia (n=2) | 0.33 [0.33, 0.33] | 0.38 [0.19, 0.56] | 0.50 [0.25, 0.75] | 0.25 [0.12, 0.38] | 0.19 [0.09, 0.28] | 2.70 [2.45, 2.95] | 1.53 [1.48, 1.58] | 2.45 [2.42, 2.48] | 0.83 [0.63, 1.02] | 0.97 [0.67, 1.27] | 0.50 [0.25, 0.75] | 0.97 [0.82, 1.11] |
| Western Pacific (n=4) | 0.88 [0.65, 1.25] | 1.38 [0.88, 1.94] | 1.00 [0.88, 1.00] | 0.41 [0.22, 0.87] | 0.61 [0.38, 0.98] | 3.00 [2.95, 3.10] | 2.11 [1.93, 2.38] | 1.87 [1.23, 2.56] | 1.79 [1.34, 2.32] | 2.24 [1.68, 2.66] | 2.50 [2.38, 2.62] | 1.71 [1.54, 1.84] |

**Note:** Scores are presented with interquartile ranges reported in brackets. LIC = low-income country; LMIC = lower-middle-income countries; UMIC = upper-middle-income countries; HIC = high-income countries; HSP = health system performance.

PLOS Digital Health

**Fig 1. Comparison of health system performance in relation to elements by World Health Organisation region.**

small subgroup sizes. The Western Pacific region (WPRO; n=4) achieved the highest median total score at 1.71 [IQR 1.54, 1.84]. This was driven by high scores in strategy (3.00 [2.95, 3.10]), implementation (2.50 [2.38, 2.62]), and health services (2.24 [1.68, 2.66]). Mid-range scores included financing (1.87 [1.23, 2.56]) and resource management (1.79 [1.34, 2.32]), while threats (0.41 [0.22, 0.87]) and opportunities (0.61 [0.38, 0.98]) remained low.

The Eastern Mediterranean region (EMRO; n=5) recorded a median total of 1.46 [1.41, 1.79]. Leading elements were strategy (3.00 [2.80, 3.00]), governance and organisation (2.43 [2.29, 2.71]), and health services (2.38 [1.62, 2.57]). Financing (2.10 [1.90, 2.10]) was also notable. However, significant gaps persisted in threats (0.25 [0.12, 0.29]), opportunities (0.60 [0.25, 0.67]), and outcomes (0.67 [0.33, 0.75]).

In the Americas (PAHO; n=8), the median total was 1.26 [1.11, 1.55]. While strategy scored highly (3.00 [2.70, 3.20]), followed by implementation (2.25 [1.31, 2.50]), other areas were weaker, including health services (1.50 [1.14, 1.89]) and financing (1.60 [1.08, 1.82]). The lowest scores were observed in threats (0.12 [0.12, 0.17]) and opportunities (0.19 [0.12, 0.38]).

The European region (EURO; n=22) had a median total of 1.14 [0.87, 1.40]. The highest scoring elements were strategy (2.40 [2.00, 3.15]) and governance (2.29 [1.43, 2.43]). Implementation (1.38 [0.54, 2.25]), financing (1.35 [0.93, 1.70]), and health services (1.33 [0.82, 1.68]) showed moderate completeness. Contextual analysis was poor, with threats scoring 0.12 [0.12, 0.34].

South-East Asia (SEARO; n = 2) recorded a median total of 0.97 [0.82, 1.11]. Despite high scores in strategy (2.70 [2.45, 2.95]) and financing (2.45 [2.42, 2.48]), most elements scored poorly, including implementation (0.50 [0.25, 0.75]), outputs (0.50 [0.25, 0.75]), and threats (0.25 [0.12, 0.38]).

The African region (AFRO; n = 4) showed the lowest median total at 0.90 [0.80, 1.02]. While strategy (2.30 [1.50, 3.05]) was relatively strong, threats scored 0.00 [0.00, 0.06]. Implementation (0.38 [0.19, 0.69]), health services (0.86 [0.71, 1.21]), and financing (0.95 [0.67, 1.23]) were also low.

This figure shows comprehensiveness scores for 45 NCVDCPs across 11 elements, stratified by WHO region. Box plots display the distribution (median and IQR) of element scores, with individual countries shown as dots and coloured by region. Element scores represent the median across constituent sub-categories.

## Income group variation

Income group comparisons are presented descriptively; all pairwise Mann-Whitney comparisons were non-significant (all p > 0.05), and the LIC group (n = 2) is insufficient for any inferential conclusion. Analysis by World Bank country income classification is presented in Fig 2. Low-income countries (LIC; n = 2) recorded a median total score of 1.39 [1.32, 1.46]. These plans recorded their highest scores in governance (2.64 [2.04, 3.25]), financing (2.55 [2.48, 2.62]), strategy (2.50 [2.15, 2.85]), and implementation (2.00 [1.50, 2.50]). The lowest scores were observed in outcomes (0.50 [0.42, 0.58]) and threats (0.31 [0.22, 0.41]).

**Fig 2. Comparison of health system performance in relation to elements by World Bank country income group.**

High-income countries (HIC; n = 27) recorded a median total score of 1.30 [1.05, 1.67]. Leading elements included strategy (3.00 [2.40, 3.20]), governance and organisation (2.29 [1.79, 2.43]), and implementation (2.00 [0.75, 2.50]). Health services (1.50 [1.14, 2.29]) and financing (1.50 [1.10, 1.95]) scored in the moderate range. Contextual threats (0.12 [0.12, 0.33]) and opportunities (0.38 [0.25, 0.71]) scored lowest.

Upper-middle-income countries (UMIC; n = 10) recorded a median total of 1.13 [0.90, 1.37]. Strategy (2.40 [2.10, 2.95]) and resource management (1.94 [0.92, 2.08]) were the highest scoring elements. Financing (1.50 [0.93, 1.90]) and implementation (1.25 [0.56, 2.19]) showed moderate coverage. Threats (0.13 [0.12, 0.34]) and opportunities (0.12 [0.12, 0.28]) were lowest.

Lower-middle-income countries (LMIC; n = 6) recorded the lowest median total at 0.90 [0.72, 1.14]. Strategy (2.60 [1.75, 3.00]) and governance (1.60 [1.42, 1.80]) were the highest scoring elements. Implementation (0.38 [0.06, 1.06]) and threats (0.00 [0.00, 0.19]) recorded the lowest scores.

This figure shows comprehensiveness scores for 45 NCVDCPs across 11 elements, stratified by World Bank income group. Box plots display the distribution (median and IQR) of element scores, with individual countries shown as dots and coloured by income group. Element scores represent the median across constituent sub-categories.

## Discussion

The validated 11-element, 69-sub-element framework constitutes a reusable policy instrument applicable to national CVD planning assessment independent of the computational pipeline described here. It was developed through systematic literature review and two-stage Delphi consensus with 42 specialists. Our analysis of 45 NCVDCPs indicates that while high-level strategic vision is often present, the financing, governance and operational architecture required to execute that vision is frequently under-specified. A median overall comprehensiveness score of 1.20/5 reveals a substantial gap between the scale of the global CVD burden and the granularity of national planning documentation.

Most plans articulate a Strategy (median 2.80) and define Governance structures (2.14); however, they frequently lack a documented grounding in local context or baseline system performance. Negligible scores for Contextual Threats (0.12) and Opportunities (0.29), combined with limited definitions of Health System Outcomes (0.67), indicate that strategies are frequently developed without documented epidemiological baselines or health-system assessments. Without a documented assessment of the current system status or external factors that might impede progress – such as economic instability or demographic shifts – even a well-articulated strategy risks remaining a signal of political intent rather than an actionable roadmap.

Among the sampled plans, the Western Pacific region recorded the highest median (1.71), with stronger documentation of Implementation and Health Services; the African region recorded the lowest (0.90), with Contextual Threats scoring 0.00. Both findings are based on n = 4 countries and should not be interpreted as representative regional assessments. Income group comparisons were non-significant across all pairwise tests, and the LIC sample (n = 2) is insufficient for inference. Descriptive patterns — including LMIC scoring lowest (median 0.90) — are reported without causal attribution. The primary finding — that comprehensiveness was low across all income groups — suggests the deficit reflects the absence of standardised planning methodology more than resource constraints, reinforcing the case for universally adaptable templates similar to those established for cancer control [19–23].

Three major implications follow from these findings. First, national CVD planning must transition from vision-based to evidence-based approaches. The divergence between strong scores for Strategy and weaker scores for Outcomes (0.67) and Objectives (0.50) indicates a disconnect between goals and the metrics required to track them. To fulfil WHA and SDG mandates and targets, CVD planning frameworks should explicitly link strategic goals to baseline data and measurable targets. This alignment is critical for tracking progress toward mortality reduction targets, yet monitoring and evaluation frameworks currently score poorly (1.75) across the sample.

Second, limited detail regarding Financing (1.44) and Contextual Analysis jeopardises plan sustainability. A comprehensive plan requires not just a budget, but an analysis of fiscal space, funding sources, and financial risk protection. The

omission of these details suggests financial planning is frequently decoupled from service delivery planning. Future planning should integrate service delivery objectives with financing and implementation architecture to build resilience against political or economic shocks.

Third, the application of PRISM demonstrates both the potential and the current limitations of AI-enabled policy appraisal. By automating the analysis of complex policy documents, PRISM enables scalable benchmarking and a dynamic global observatory for CVD policy. However, validation against human review reveals important nuances in its application. While PRISM achieved exact agreement with human experts in 43.7% of comparisons and fell within a one-point margin in 68.0% of cases, ordinal agreement statistics (Spearman $\rho = 0.061$, weighted $\kappa = 0.049$, ICC $= 0.046$), confirm that PRISM reliably identifies structural presence but does not replicate expert qualitative depth assessment. Bias was score-dependent rather than uniformly generous: PRISM over-scored when reviewers found no substantive content (mean difference +0.96 at human score 0) but under-scored substantially when reviewers identified genuine policy depth (−0.93 to −4.00 at scores 2–5), consistent with a keyword-trigger effect, where the model rewards stated intent over demonstrated specificity.

This pattern was reflected at the element level: PRISM performed best on elements requiring structural detection (Element 4, exact agreement 72.9%; Element 5, 72.9%), and worst on elements requiring qualitative depth assessment (Element 6, 16.7%; Element 7, 19.0%; Element 8, 37.3%). Representative cases illustrate both failure modes: PRISM assigned scores of 4–5 for Vision and Financing sub-elements in Ghana, Myanmar, and the United States where human reviewers scored 0, rewarding aspirational language without implementation architecture; conversely, in India, PRISM scored 0 across Governance, Outputs, and Health Services sub-elements where reviewers assigned 3–4, failing to integrate content across document sections or apply contextual inference. Full disagreement cases ($|diff| \geq 3$, n = 84) are provided in S3 Table.

This bidirectional pattern reflects score compression — PRISM's output clusters in the low-to-middle range, missing both the complete absence and the substantive depth that expert reviewers reliably distinguish. PRISM is therefore validated for structural benchmarking of documented plan content. It should not be used as a fine-grained ordinal scoring instrument equivalent to expert judgement, and serves best as a screening tool to augment, rather than replace, expert review.

Future iterations should explore few-shot prompting — providing PRISM with scored exemplars prior to evaluation — and stricter criteria for awarding scores above 0; both represent tractable approaches to reducing over-scoring of nominal policy language.

These findings must be interpreted in light of significant limitations.

First, PRISM evaluates document content, which may not reflect implementation reality or policies detailed in separate, unlinked documents (e.g., national budgets). Thus, low scores indicate a lack of integration within the primary CVD strategy, not necessarily government inaction. Additionally, formal analysis of scoring reliability by document language or format was not conducted. RAG-based retrieval may also perform less reliably on contextual content dispersed across narrative text than on explicitly structured policy sections, potentially contributing to the very low observed scores for threats and opportunities. Plans were processed in any language using a multilingual ingestion agent; whether reliability varied systematically by language or document structure cannot be determined from the current validation sample.

Second, the sample is geographically and economically skewed: 22 of 45 countries are European, 27 are high-income, and low-income (n = 2), African (n = 4), and South-East Asian (n = 2) contexts are substantially under-represented; findings are most generalisable to high-income, European settings. Countries without accessible standalone CVD plans — including China — could not be included, which is itself consistent with the study's central finding on the scarcity of dedicated CVD planning. Expanding coverage across Africa, South-East Asia, and Latin America is the primary direction for future work.

Finally, PRISM's validated scope is structural: it identifies whether defined policy architecture is present, not whether that architecture is sufficient, feasible, or of high quality.

| Contributor Name | Role/Affiliation | Participation in Stage One | Participation in Stage Two |
|---|---|---|---|
| Dr Laurie Whitsel | National Vice President Policy Research and Translation, American Heart Association | Yes | Yes |
| Professor Angela Koh Su-Mei | Senior Consultant National Heart Centre Singapore Department of Cardiology and Director of Cardiovascular Ageing & Longevity Program at the National Heart Centre Singapore | Yes | No |
| Professor Jaime Miranda | Professor and Head of School, University of Sydney Faculty of Medicine and Health, Sydney School of Public Health | Yes | No |
| Professor Carlos Mendivil Anaya | Professor School of Medicine University of the Andes Bogota Columbia | Yes | No |
| Professor Rodrigo Carrillo Larco | Assistant Professor, Rollins School of Public Health, Emory University | Yes | No |
| Dr Nasirumbi Magero | Consultant Epidemiologist, Ministry of Health Kenya | Yes | Yes |
| Professor Fernanda de Carvalho | Representative to the World Heart Federation Advocacy Committee, Institutional and International Relations, Public and International Affairs Instituto Lado a Lado Pela Vida | Yes | Yes |
| Chinonso Amanda Ugwu | Assistant lecturer at University of Medical Sciences, Ondo | Yes | Yes |
| Urlish Kleyber Marroquin Marroquin | Pan American Health Organisation, Lima, Peru | Yes | Yes |
| Dr Wallace Odiko-Ollennu | Program Manager Odiko-Ollennu Programme Manager Non-Communicable Disease Control Programme Ghana Health Service | Yes | Yes |
| Dr Ad Adams | Executive Director Stroke Association Support Network Ghana, Vice-Chairman Ghana NCD Alliance | Yes | Yes |
| Shawna Novak | Department of Global Health and Social Medicine, Harvard Medical School; and Ariadne Labs, Harvard T.H. Chan School of Public Health, Adjunct Professor University of Toronto | Yes | Yes |
| Dr Mayanak Dalakoti | Cardiologist, National University Heart Centre, Singapore; Prevention & Cardiometabolic Health, Asian Pacific Society of Cardiology. | Yes | No |
| Dr Bolanle Banigbe | Technical Director for Global Hypertension Control, Resolve to Save Lives | Yes | Yes |
| Dr Shafika Abrahams-Gessel | Research Scientist, Center for Health Decision Science Harvard T.H. Chan School of Public Health | Yes | Yes |
| Dr Sadhan Das | DBT/Wellcome Trust India Alliance Intermediate Fellow, Assistant Professor Indian Institute of Science Education and Research Mohali | Yes | No |
| Nida Wasim | GBD Collaborator, Institute for Health Metrics and Evaluation | Yes | Yes |
| Dr John T Figi | Postdoctoral Research Fellow, Harvard Health Systems Innovation Lab | Yes | Yes |
| Dr Alem Aminu Osman | Postdoctoral Research Fellow, Harvard Health Systems Innovation Lab | Yes | Yes |
| Professor Kai Kappert | Director Institute of Laboratory Medicine, Clinical Chemistry and Pathobiochemistry Charité – Universitätsmedizin Berlin | Yes | No |
| Patricia Codyre | Rapporteur, Directorate-General for Research and Innovation (DG RTD) European Commission, Brussels, Belgium | Yes | Yes |
| Isabela Castro | Executive Director, Instituto Isabela Castro; and Innovation Advisor & Fellow, Institute for Healthcare Improvement | Yes | Yes |
| Dr Joy Aifuobhokhan | Chief Operations Officer, AwaDoc; and Digital Health Lead, Lakeshore Cancer Center | Yes | Yes |

## Supporting information

**S1 Text. Large language model pipeline architecture overview.**
(DOCX)

**S2 Text. Delphi process stage one questionnaire quantitative scoring by element.**
(DOCX)

**S3 Text. Delphi process stage one questionnaire qualitative feedback by element.**
(DOCX)

**S4 Text. Delphi process stage two qualitative feedback summary by key themes.**
(DOCX)

**S5 Text. Quantitative analysis of national cardiovascular disease control plans across all sub-elements.**
(DOCX)

**S1 Table. National cardiovascular disease control plan scoring rubric.**
(DOCX)

**S2 Table. Sensitivity analysis of scoring aggregation methods.**
(DOCX)

**S3 Table. Large disagreements between LLM and human reviewer scores (|difference| ≥ 3).**
(DOCX)

## Acknowledgments

Members of CVD control collaborative

**Note:** Stage One: 42 completed questionnaires (participants not listed elected to remain anonymous), Stage Two: 16 participants (participant not listed elected to remain anonymous).

**Human Review of LLM**

Completed by Dr Aminu Osman Alem, Maia Cullen, Brooke Forde from the Harvard University Health Systems Innovation Lab

## Author contributions

**Conceptualization:** Hugh Pearson, Caleb J. Kumar, Rifat Atun.

**Data curation:** Caleb J. Kumar.

**Formal analysis:** Hugh Pearson, Caleb J. Kumar.

**Investigation:** Hugh Pearson.

**Methodology:** Caleb J. Kumar, Che L. Reddy.

**Project administration:** Hugh Pearson.

**Resources:** Estella Rose LeBlanc.

**Software:** Caleb J. Kumar.

**Supervision:** Rifat Atun.

**Validation:** Caleb J. Kumar.

**Visualization:** Caleb J. Kumar.

**Writing – original draft:** Hugh Pearson.

**Writing – review & editing:** Hugh Pearson, Che L. Reddy, Estella Rose LeBlanc, Rifat Atun.

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
