## [Decision Letter · Decision Letter 0]

3 Mar 2026

PDIG-D-26-00060A global analysis of national cardiovascular disease control plans using a multi-agent artificial intelligence modelPLOS Digital Health Dear Dr. Rifat Atun, Thank you for submitting your manuscript to PLOS Digital Health. After careful consideration, we feel that it has merit but does not fully meet PLOS Digital Health's publication criteria as it currently stands. Therefore, we invite you to submit a revised version of the manuscript that addresses the points raised during the review process. Please submit your revised manuscript by May 02 2026 11:59PM. If you will need more time than this to complete your revisions, please reply to this message or contact the journal office at digitalhealth@plos.org.  Please include the following items when submitting your revised manuscript:* A letter that responds to each point raised by the editor and reviewer(s). You should upload this letter as a separate file labeled 'Response to Reviewers'. This file does not need to include responses to any formatting updates and technical items listed in the 'Journal Requirements' section below.* A marked-up copy of your manuscript that highlights changes made to the original version. You should upload this as a separate file labeled 'Revised Manuscript with Track Changes'.* An unmarked version of your revised paper without tracked changes. You should upload this as a separate file labeled 'Manuscript'. If you would like to make changes to your financial disclosure, competing interests statement, or data availability statement, please make these updates within the submission form at the time of resubmission. Guidelines for resubmitting your figure files are available below the reviewer comments at the end of this letter. We look forward to receiving your revised manuscript. Kind regards, Cleva Villanueva, M.D., Ph.D.Academic EditorPLOS Digital Health Cleva VillanuevaAcademic EditorPLOS Digital Health Leo Anthony CeliEditor-in-ChiefPLOS Digital Healthorcid.org/0000-0001-6712-6626  **Journal Requirements:** 

1. Please provide a detailed online Financial Disclosure statement. This is published with the article. It must therefore be completed in full sentences and contain the exact wording you wish to be published.

a) State the initials, alongside each funding source, of each author to receive each grant. For example: “This work was supported by the National Institutes of Health (####### to AM; ###### to CJ) and the National Science Foundation (###### to AM).”

For more information, please go to our submission guidelines:

https://journals.plos.org/digitalhealth/s/submission-guidelines#loc-financial-disclosure-statement

2. Please ensure that the funders and grant numbers match between the Financial Disclosure field and the Funding Information tab in your submission form. Note that the funders must be provided in the same order in both places as well.

3. Please update your online Competing Interests statement. If you have no competing interests to declare, please state: "The authors have declared that no competing interests exist."

4. Thank you for submitting your data to a repository. We have been unable to access the data using the link/accession number you provided. Please contact the repository to ensure that the links and accession numbers are valid. If a link or accession number needs updating, please provide a revised Data Availability Statement including the information.

For further assistance, you may go to: http://journals.plos.org/digitalhealth/s/data-availability

5. Please provide separate main figure files in .tif or .eps format only and ensure that all files are under our size limit of 10MB.

For more information about how to convert your figure files please see our guidelines: https://journals.plos.org/digitalhealth/s/figures

 **Additional Editor Comments (if provided):** This is an important and policy-relevant study with a well-developed framework and an innovative AI-based approach. However, substantial methodological concerns must be addressed before the manuscript can be considered for publication.

1. Validation and Statistical Concerns

The reported Pearson correlation between PRISM and blinded human review (r = 0.048) indicates essentially no linear agreement. This critically weakens claims regarding model validity.

Moreover, the scoring scale is ordinal (1–5), yet analyses treat it as continuous. The authors must:

• Use appropriate statistics for ordinal data (e.g., Spearman correlation, weighted kappa, ICC).

• Conduct a formal bias analysis to address the reported +0.28 “generous auditor” effect.

• Clarify the interpretation of the scoring scale, including what constitutes a “good” score.

Without stronger validation evidence and proper statistical treatment, conclusions about AI reliability are not sufficiently supported.

2. Selection Bias and Limited Global Representation

The country sample shows substantial imbalance:

• 22/44 countries are European.

• Very few low-income countries are included.

• Key global actors are missing.

As a result, the global median score is disproportionately influenced by European data. The conclusion that low-income countries perform similarly to high-income countries is statistically unreliable given the extremely small number of low-income cases.

To support global claims, the study must either:

• Expand representation of low- and middle-income countries (particularly from Africa, Southeast Asia, and Latin America), or

• Substantially temper conclusions and explicitly acknowledge structural sampling bias.

3. Model Interpretation and Transparency

Given the weak correlation with human scoring, the manuscript must more clearly distinguish between:

• Structural identification of policy components, and

• Qualitative evaluation of policy depth or feasibility.

A concrete example of AI–human scoring disagreement would strengthen transparency.

In its current form, the manuscript presents promising methodology but insufficient statistical robustness and limited external validity. These issues must be addressed in a major revision.**Reviewers' Comments:** Reviewer's Responses to Questions

**Comments to the Author**

1. Does this manuscript meet PLOS Digital Health’s publication criteria? Is the manuscript technically sound, and do the data support the conclusions? The manuscript must describe methodologically and ethically rigorous research with conclusions that are appropriately drawn based on the data presented.

Reviewer #1: Partly

Reviewer #2: Yes

Reviewer #3: Yes

2. Has the statistical analysis been performed appropriately and rigorously?

Reviewer #1: Yes

Reviewer #2: Yes

Reviewer #3: Yes

3. Have the authors made all data underlying the findings in their manuscript fully available (please refer to the Data Availability Statement at the start of the manuscript PDF file)?

Reviewer #1: Yes

Reviewer #2: Yes

Reviewer #3: Yes

4. Is the manuscript presented in an intelligible fashion and written in standard English?

Reviewer #1: Yes

Reviewer #2: Yes

Reviewer #3: Yes

5. Review Comments to the Author

Reviewer #1: This is an important and well structured study with clear policy relevance. To strengthen the manuscript, please clarify the intended scope of validation, add diagnostic analyses explaining disagreement with human scoring, align statistical reporting with the ordinal nature of the rubric, and demonstrate robustness of aggregation choices. Minor corrections to labeling and clarification of scoring interpretation will further improve clarity.

Reviewer #2: This is an interesting and novel implementation of an LLM to assist a large volume of CVD policy documents based on a Validated policy framework. The comprehensive validation of the framework provide confidence in the underlying method of policy assessment and shows value for future work in this area. The supplementary document shows a comprehensive implementation of an LLM-agent pipeline with good standardisation and guardrails to ensure consistent and reliable output. The results show a relatively poor score based on the framework applied and provide good discussion on which areas of the framework need improvement, overall and from a regional perspective. The agreement between the LLM output and expert review seemed poor, which does put into question how reliable these results are, however this is adequately discussed and the findings that an LLM might be able to structural identify the presence of policy but not evaluate its quality is an important one.

I had two small points to question.

First, The final scoring scale of 1 to 5, it's unclear what a "good" score is of what these points mean. the highest scores in strategy, 2.8, being a bit over half seemed bad to me, but this discussion notes this result as "successful" (line 309). Some clarify about what the score means and what is "good" would be helpful to interpret the results.

Second, there seems to be some formatting errors in Table 3. The median score of 2.8 is for element 6 "threats" but the results text says this score is for strategy, also the table "Cancer strategy" not "CVD". and the order of the elements in Table 1 and Table 3 are not consistent.

Reviewer #3: This study analyzes national cardiovascular disease control plans across 44 countries using a multi-agent AI model. It addresses a significant gap in the structured comparison of global health policies. Furthermore, the research demonstrates the practical potential of Large Language Models for processing and analyzing complex policy documents. However, some questions still remain to be answered.

1.The study’s sample may have some selection bias. Specifically, the number of low-income countries included is very small, and the regional distribution is uneven—with 22 countries in Europe, but only 2 in Southeast Asia and 4 in Africa. This means the global median score is heavily skewed by the European data. Furthermore, data from key representative countries like China and the US is missing. Drawing the conclusion that "low-income countries perform similarly to high-income ones" based on only two data points is statistically unreliable and could be misleading.

2.The study shows that the Pearson correlation between PRISM and human blind reviews is only r=0.048. In statistics, this is essentially the same as having no linear correlation. Additionally, the authors mention a "generous auditor" effect, where the AI scores are an average of 0.28 points higher than human scores. It is worth exploring whether this systematic bias can be fixed by using "few-shot prompting" or by introducing stricter rules for negative evaluations.

3.There is also the question of a "keyword-trigger" effect. Does the AI give high scores simply because it sees certain words? In contrast, human experts evaluate the logical depth and feasibility of the content. To illustrate this, the paper should include an extreme comparison—specifically, a case where the AI gives a high score while a human reviewer gives a low one.

4.The research framework is adapted from a 2008 cancer control plan, even though CVD and cancer have very different management models. While the paper mentions that the framework was updated using the Delphi method, Table 1 still seems to show traces of the original version. For example, Element 3 includes "Cancer Surveillance Systems." Please verify if the use of the word "cancer" here is simply a typo.

5.While this study shows the huge potential of AI for large-scale policy analysis, its effectiveness as a qualitative tool should still be interpreted with caution.

6. PLOS authors have the option to publish the peer review history of their article (what does this mean?). If published, this will include your full peer review and any attached files.

**Do you want your identity to be public for this peer review?** For information about this choice, including consent withdrawal, please see our Privacy Policy.

Reviewer #1: No

Reviewer #2: **Yes:** Calum Nicholson

Reviewer #3: No

  **Figure resubmission:** While revising your submission, we strongly recommend that you use PLOS’s NAAS tool (https://ngplosjournals.pagemajik.ai/artanalysis) to test your figure files. NAAS can convert your figure files to the TIFF file type and meet basic requirements (such as print size, resolution), or provide you with a report on issues that do not meet our requirements and that NAAS cannot fix.

After uploading your figures to PLOS’s NAAS tool - https://ngplosjournals.pagemajik.ai/artanalysis, NAAS will process the files provided and display the results in the "Uploaded Files" section of the page as the processing is complete. If the uploaded figures meet our requirements (or NAAS is able to fix the files to meet our requirements), the figure will be marked as "fixed" above. If NAAS is unable to fix the files, a red "failed" label will appear above. When NAAS has confirmed that the figure files meet our requirements, please download the file via the download option, and include these NAAS processed figure files when submitting your revised manuscript. **Reproducibility:** To enhance the reproducibility of your results, we recommend that authors of applicable studies deposit laboratory protocols in protocols.io, where a protocol can be assigned its own identifier (DOI) such that it can be cited independently in the future. Additionally, PLOS ONE offers an option to publish peer-reviewed clinical study protocols. Read more information on sharing protocols at https://plos.org/protocols?utm_medium=editorial-email&utm_source=authorletters&utm_campaign=protocols

---

## [Decision Letter · Decision Letter 1]

5 May 2026

A global analysis of national cardiovascular disease control plans using a multi-agent artificial intelligence model

PDIG-D-26-00060R1

Dear Dr. Atun,

We are pleased to inform you that your manuscript 'A global analysis of national cardiovascular disease control plans using a multi-agent artificial intelligence model' has been provisionally accepted for publication in PLOS Digital Health.

Best regards,

Cleva Villanueva, M.D., Ph.D.

Academic Editor

PLOS Digital Health

**Additional Editor Comments (if provided):**

After carefully reviewing the revised version of the manuscript, along with the comments from the reviewers of the original submission—who have now approved the revised version—I am pleased to inform you that the manuscript has been accepted for publication. The authors have adequately addressed the reviewers’ comments, and the manuscript meets all the requirements of PLOS Digital Health

**Reviewer Comments (if any, and for reference):**

Reviewer's Responses to Questions

**Comments to the Author**

1. If the authors have adequately addressed your comments raised in a previous round of review and you feel that this manuscript is now acceptable for publication, you may indicate that here to bypass the “Comments to the Author” section, enter your conflict of interest statement in the “Confidential to Editor” section, and submit your "Accept" recommendation.

Reviewer #2: All comments have been addressed

Reviewer #3: All comments have been addressed

2. Does this manuscript meet PLOS Digital Health’s publication criteria? Is the manuscript technically sound, and do the data support the conclusions? The manuscript must describe methodologically and ethically rigorous research with conclusions that are appropriately drawn based on the data presented.

Reviewer #2: Yes

Reviewer #3: Yes

3. Has the statistical analysis been performed appropriately and rigorously?

Reviewer #2: Yes

Reviewer #3: Yes

4. Have the authors made all data underlying the findings in their manuscript fully available (please refer to the Data Availability Statement at the start of the manuscript PDF file)?

Reviewer #2: Yes

Reviewer #3: Yes

5. Is the manuscript presented in an intelligible fashion and written in standard English?

Reviewer #2: Yes

Reviewer #3: Yes

6. Review Comments to the Author

Reviewer #2: Thank you for the thoughtful responses to this review, the updated analysis is much more robust now, with clearly outlined measures and appropriate statistical tests. This update reveals a clearer insight into the agreement between expert and LLM and allows the paper to provide a more significant contribution to the application of this methodology. The updated discussion around about how disagreement is occurring, and noting factors such as the "score compression" and the limited ability of the LLM to identify structural presence only, and not quality, are important insights. This does limit the overall conclusions about the quality of national plans when reviewed by LLMs, but as an overall exploration of this technology and this specific implementation, this is a worthwhile contribution.

Reviewer #3: Accept!

7. PLOS authors have the option to publish the peer review history of their article (what does this mean?). If published, this will include your full peer review and any attached files.

**Do you want your identity to be public for this peer review?** For information about this choice, including consent withdrawal, please see our Privacy Policy.

Reviewer #2: No

Reviewer #3: No
